# "Some believe those who say they can cure it" perceived barriers to antiretroviral therapy for children living with HIV/AIDS: Qualitative exploration of caregivers experiences in tamale metropolis

**Gideon Awenabisa Atanuriba**[1]*, **Felix Apiribu**[2], **Veronica Millicent Dzomeku**[2], **Philemon Adoliwine Amooba**[2], **Adwoa Bemah Boamah Mensah**[2], **Richard Adongo Afaya**[3], **Timothy Gazari**[3], **Timothy Tienbia Laari**[2,4], **Moses Haruna Akor**[5], **Linda Abnory**[6]

1 Tamale Central Hospital, Ghana Health Service, Tamale, Northern Region, Ghana, 2 Department of Nursing, Faculty of Allied Health Sciences, College of Health, Sciences, Kwame Nkrumah University of Science and Technology, Kumasi, Ghana, 3 School of Nursing and Midwifery, University for Development Studies, Tamale, Ghana, 4 Presbyterian Primary Health Care (PPHC), Bolgatanga, Ghana, 5 Nurses and Midwifery Training College-Damongo, Savanna Region, Ghana, 6 Madonna Health Center, Ejisu-Besease, Ghana

* atanuriba@gmail.com

## Abstract

### Background

HIV/AIDS is now a chronic disease, as adherence to anti-retrovirals impacts positively on the quality as well as expectancy of life. However, there exist multifaceted barriers to treatments for which children are most disadvantaged. Since Ghana subscribed to the "treat all" policy less percentage (25.5%) of children (2–14 years) living with HIV/AIDS have been enrolled on the antiretroviral program compared to other categories of the population by 2019. At present no study has explored these barriers to children living with HIV/AIDS enrollment and adherence. This study aims to explore the perceived barriers of caregivers of children living with HIV/AIDS in the Tamale Metropolis.

### Methods

We used descriptive phenomenology to explore the phenomena. Caregivers were purposively selected and interviewed till information became repetitive at the ninth (9th) caregiver. A semi-structured interview guide was used to collect data through face-to-face in-depth interviews which were audio recorded. The interviews lasted an average of 47 minutes. Audio interviews were transcribed verbatim (English) and translated back-to-back (Daghani) before analysis was done manually according to Collaizi's seven-step approach. We used the Guba and Lincoln guidelines to ensure the rigour of the study and its findings. Results are presented in themes and supported with quotes.

**Data Availability Statement:** Data is available upon request and has been attached as a supporting document in this submission.

**Funding:** The authors received no specific funding for this work.

**Competing interests:** The authors have declared that no competing interests exist.

**Abbreviations:** AIDS, Acquired Immune-Deficiency Syndrome; ART, Antiretroviral therapy; ARVs, Anti-retrovirals; CD4, Cluster of Differentiation 4; CLWHA, Children Living with HIV/AIDS; HAART, Highly Active Anti-Retroviral Therapeutic; HIV, Human Immune Virus; PLWHA, People Living with HIV/AIDS; STI, Sexually Transmitted Infections; WHO, World Health Organization.

## Results

Six themes emerged from the analysis of the caregivers' transcripts; (1) denial of HIV/AID diagnosis, (2) stock-outs and privacy at the clinic, (3) busy schedule and poor support, (4) ignorance and alternative herbal cure, (5) stigma and discrimination, (6) transportation and distance.

## Conclusion

Perceived barriers are multi-dimensional and encountered by all PLWHA, especially children. These barriers could derail the gains of HIV/AIDS interventions among children. Adherence counselling among caregivers alongside campaigns among faith and herbal healers are of grave concern to reduce myths of cure.

## Introduction

HIV/AIDS has gained a manageable chronicity status just a little over a decade after the medical technological breakthrough with Highly Active Anti-Retroviral Therapeutic (HAART). As many more infected are having their quality and expectancy of life-improving [1]. However, the effects of one of the most dared diseases of history have exerted systematic and structural effects in many resource-constrained countries especially those in Africa [2]. Visible social, economic and political connotations of HIV/AIDS have had profound direct or indirect effects on every household in the sub-Saharan region where its prevalence is the highest [1, 2].

For pediatric infections, vertical transmission during pregnancy, labour and delivery, or through breastfeeding is most common [3]. However, with AIDS free-generation agenda by the World Health Governing Body-World Health Organization (WHO) zealous target code christened 90-90-90 by 2020 has not achieved its goals globally. Signifying the aim of ensuring that 90% of all people living with HIV will know their status, 90% of all people diagnosed will receive sustained ART and 90% of all PLWHA receiving ART have viral suppression has been fulfilled [4]. Interventions have garnered increased awareness, voluntary counselling, access to ARVs, and stopping stigmatization among others. These interventions have slowed infections and optimal health of affected individuals [5, 6]. However, pieces of evidence clearly show that with voluntary counselling and testing agenda, more people are knowing their status but fewer of them are enrolling on the ART program and achieving viral suppression as envisaged [7].

After careful empiric research "treat all" policy by WHO was instituted in September 2016 where ARVs were to be provided for all PLWHA regardless of the levels of Cluster of Differentiation 4 (CD4) count. A move away from prior guidelines to ART where interventions with cut-off points by CD4 count. By October same year Ghana ascribed to the policy [8]. This meant HIV and AIDS care required not just emphasis on improving prevention interventions but scaling up measures to ensure the availability of ART and viral load testing services in Ghana. About five years of adopting the "treat all" policy, though there has been steady progress, children are still at the mercy of several impediments to proper access and adherence to ARVs [9, 10].

On the global front, the 90-90-90 agenda has achieved 81%-67%-59% at a prevalence of 0.7% (38.0million) [11] whereas, Ghana with a prevalence of 1.69 (334,713 PLWHA), has since achieved 58%-77%-68% by 2019. In Ghana, there are about 25,955(8%) CLWHA (0–14 years), with an estimated 2,972(15%) new infections and 2,441 (18%) deaths [9]. ART coverage

has disproportionately been unfavourable to children who are powerless and have their ability to benefit from ARVs in the hands of their caregivers. Data shows 46.55% ART coverage among adults (15+ years), 53.76% women (15+ years) and just 25.55% children (0-14years) living with HIV [9, 10].

Evidence suggests optimal and strict adherence of 95% and above is required over the long term to ensure the full benefits of the treatment [12]. Large-scale surveys have buttress reports that show high levels of positive outcomes of ARVs adherence should range between 80%-95% depending on the varying levels of ARV potency [13]. However, they report about 23% of Africans on ART were achieving less than 80% proper adherence which negatively impacts their prognosis.

Proper enrollment and adherence to ART have been documented to; ensure suppression of viral replication, progressively halt the destruction of CD4 cells, prevent viral resistance, boost immune rebuilding of severely immune-suppressed, increase life expectancy, improved quality of life, decrease side or adverse effects of ARVs, and slow disease progression [14–16].

Adherence to ART among people living with HIV/AIDS (PLWHA) is multifaceted and requires; a proper understanding of ART and HIV, conscious efforts to comply with adherence counselling, education, changing lifestyle, proper nutrition, presence of psychosocial support, active supportive care and availability of a considerate home caregiver [12]. Implying an inter-related coordinated effort involving PLWHA, their home caregivers and formal health caregivers is considered crucial [17].

Evidence suggests poor adherence to ART regimens is highly associated with low therapeutic success, higher viral loads, increased frequency of hospital visits, cost of treatment, longer stays on admission, viral resistance, disabilities and even deaths [16, 18]. Studies on barriers to enrollment and adherence include; stigma, food insecurity, troubling side effects, poor knowledge of ART, stress, lack of support, pill burden, substance abuse, and depression among others [13, 19]. Other evidence in South Africa [20] has identified stock-outs and shortages as a barrier to ART adherence. Over time HIV/AIDS diagnosis has been symbolized by death sentence and ugly associated with morality, personal failures and promiscuity [21] admonished. They explained this has intensified the shame, guilt and stigma associated with HIV/AIDS serving as a barrier for PLWHA to willingly seek HIV care and adhere to treatment. Similarly, in Uganda where [4] qualitatively explored these barriers; treatment for opportunistic infections and antiretroviral drugs were most reported to be inadequate. As many PLWHA are afraid to start ARVs due to shortages, distance, staffing problems/ shortage, persistent stigma, and lack of social and economic support initiatives that will favour retention.

Facilitators of ART adherence explored in Nigerian PLWHA reported 85.53% satisfied with the providers' interpersonal skills, and environment at the centre (77.63%) [22]. For children less than 15 years, globally 91% of the infected about 3.2 million children reside in sub-Sahara Africa by 2015. The gendered nature of caring and nurturing has shifted enormous strain on vulnerable females who assume roles of caring for children living with HIV/AIDS (CLWHA). The Guidelines for Antiretroviral Therapy in Ghana recognize the unique role of guardians or caregivers as crucial to the successful initiation and maintenance of ART among children [8]. The document avers the lack of reliable caregivers as a setback to the health and welfare of CLWHA.

'No child should die due to lack of access to treatment' USAID declared [19]. This agenda together with WHO seeks to ensure everyone child living with HIV has access to life-saving medication in the form of ARVs. To better understand this phenomenon among children, home caregivers are crucial, because they are pivotal in the health-seeking choices of these vulnerable populations. In Ghana, prior studies on the barriers to ART access occurred before the "treat all" policy and often in Southern Ghana [23, 24]. To the best of our knowledge, no study

has explored the perceived barriers to ART enrollment and adherence among children in Ghana. This qualitative exploration seeks to provide light on the perceived barriers to ART enrollment and adherence among children living with HIV/AIDS from their home caregivers.

## Materials and methods

### Design, setting, and population

We used a descriptive phenomenological approach to explore the perceived barriers to ART enrollment and adherence among home caregivers of CLWHA. This approach was used because it seeks to describe the life experiences of people about the phenomena. This type of phenomenology was first described by Husserl, referring to the meaning and essence people have about a phenomenon is described while making sure researcher(s) presuppositions are bracketed [25]. The study, therefore, was undertaken with recourse to the descriptive phenomenological philosophical assumptions of; bracketing, a less rigid approach to methods as it emerges in the course of the study, the intentionality of consciousness, and refusal of subjectivity and objectivity dichotomy since the approach relies on both [25, 26].

The study was conducted in the Tamale metropolis, the capital of the northern region by extension the whole of northern Ghana [27]. The metropolis is the third-largest urban area in Ghana. It's inhabited by 233,252 people, representing 9.4% of the region's population with 80.8% being urban dwellers. About 60.1% and 60% are literate and economically inactive respectively [27]. According to Ghana AIDS Commission in 2019 [9], the region has an HIV prevalence of 0.31% (3,697) while the metropolis stands at a prevalence of 0.68%.

Three hospitals were purposively selected for this study due to the presence of structured ART Clinics. These facilities are comprised of a tertiary (Tamale Teaching Hospital) and two secondary level hospitals (Tamale West Hospital and Tamale Central Hospital). They served as referral centres for the five northern regions and neighbouring countries such as Togo, Ivory Coast and Burkina Faso. The hospitals serve as training centres for students and provide in-patient services as well as out-patient care services.

Home caregivers of CLWHA who attended ART clinics in the metropolis were purposively sampled as the population for the study. Home caregivers refer to informal active people who care for the child at home. They ensure the spiritual, financial, physical and emotional as well as psychological needs of the children are met. Pivotal among their roles is to ensure children get enrolled on the ART program, adhere to clinic appointments for collecting ARVs supplies and adherence to the therapeutic regimen [8]. Their exposition of the phenomena perceived barriers because they were asked, what are some of the reasons that prevent other caregivers from bringing their CLWHA to the ART clinics for treatment? Caregivers aged 18–80 years actively caring for children 2–14 years who provided at least 6 months of continual care were included in this study. These caregivers involved in the study were those who could speak English or Daghani and care for children who are not newly diagnosed with HIV/AIDS. This was to take away the expressions of newly diagnosed (less than six months) who may express extreme experiences because of the acute nature of being exposed to the phenomena.

### Sampling and sample size determination

We used purposive sampling to select caregivers from the target population. The sampling technique was used because of its wide relevance and usage among qualitative researchers and its ability to select well-informed participants with vast experiences with the phenomena [28]. As such, we used criterion purposive sampling where the inclusion and exclusion criteria were followed strictly.

The sample size was determined by data saturation. A point at which data became redundant and interviews were not yielding new information as it became repetitive. To ascertain saturation and the need to end interviews, they were analyzed concurrently and iteratively with data collection. Data became saturated at the ninth (9th) caregiver interview. As we recruited throughout the three ART clinics during specific scheduled ARV appointment days in the metropolis. However, whenever a prospective participant reports to a clinic where the team was not there, the Unit Head calls and time is made to meet the prospect. First interviews were analyzed carefully and the themes that emerged from those were then followed up in the subsequent interviews till saturation was achieved.

## Instrument

We constructed a semi-structured interview guide based on the specific objectives and literature. The instrument consists of two major parts. Section-A, collected biographic data of both the CLWHA (age, sex, level of education, and duration of being on ART) and the caregiver (age, sex, relations, occupation, and HIV status). Section-B, entailed questions about caregivers' perceived barriers to ART enrollment and adherence. Probes about perceived barriers bothered on; individual caregiver barriers, facility/hospital level, and community/societal factors.

## Data collection procedure

The varied biographic background (age, sex, and religion) of the caregivers did not favour focused group discussion. Also, conducting a focused group discussion would have been overwhelmingly difficult due to logistical and socio-cultural constraints. Bearing that participants lived long distances from the facilities and had different appointment dates, bringing them together for group interviews would have been extremely difficult. More so, getting participants to openly talk about their disease condition is difficult, especially about HIV/AIDS which is stigmatized. As such face-to-face in-depth individual interviews were conducted. From September to November, 2019, the lead author and a language expert in Daghani conducted the interviews at caregivers' places and times of convenience at the hospital or home of caregivers. Caregivers were introduced to the study during their ARV collection appointment days at the clinic while awaiting or after consultation. Those who could read and write were given a Participant's Information Leaflet to acquaint themselves with the study while those who could not be taken through for a fair understanding of the protocols. This allowed them to understand the; purpose, benefits, risk, voluntary consent and withdrawal among others. They were then taken through the consent process by thump printing or signing and verified by a witness before the interviewer who also signed appropriately to recruit. However, prospective participants were also informed of the study by link nurses at the various ART clinics. Contacts are then made to the lead author to recruit. Seven interviews were conducted in English by the lead author while two were conducted in Daghani by a language expert. As well eight (8) of the interviews were conducted in the hospital in a consulting room in the presence of the interviewer alone and one at the home of the caregiver.

During interviews, counsellors at the ART clinic were informed about the interviews and the possible referral for counselling should a participant breakdown. The interviews lasted about 30–60 (average of 47) minutes, were audio-recorded, and complimentary notes were made of the non-verbal communications to enrich the data. After the interviews, debriefs were made to ensure the session has not affected the participants before disengaging. They were given a bar of soap and biscuits as reciprocity for time spent with the researchers.

## Data management, analysis and rigour

The audio reordered interviews were listened to several times before verbatim transcription was made. The English interviews by the lead author while the Daghani ones were transcribed back-to-back by two Daghani language experts to ensure meaning were not lost before translation finally to English. The biographic data were also entered into a word document, collated and named. Transcribed interviews and biographic data were then given familiar file naming and stored on a pen drive and Personal Computer (PC) under password and only accessible to the research team.

Data analysis was conducted by GAA, TG and TTL manually according to the ideals of Collaizi analysis of phenomenological interviews, concurrently during data collection. We followed the seven iterative steps of Collaizi below Fig 1 to analyze the data.

During the presentation of the results, we used quotations from the participants to make emphasis and used false names to ensure confidentiality and anonymity.

To ensure the quality and trustworthiness of the study, Lincoln and Guba's [28] dimensions were used. As such peer debriefs, member checks (credibility), audit trail (dependability) and

(1) we listened to the audiotapes severally, transcribed and read through them on numerous occasions to acquaint ourself with the overwhelming meaning

(2), from each of the transcribes we identified and listed all the significant information

(3), meanings were then created from all the listed significant statement

(4), the meanings created were organized into themes and/or clusters

(5), an exhaustive description and essence of the phenomena of perceived barriers to ART enrollment and adherence were then made by integrating the themes

(6) bearing from the setting and context of perceived barriers of ART enrollment and adherence definitive unequivocal statement about the understanding of this phenomena were made. Thus, becoming the final meanings.

(7) the lead author then made a final validation step of sending randomly five (5) of the final analysis to the caregivers for verification.

**Fig 1. Collaizi steps used to analyze data.**

bracketing of prior pre-suppositions, inter-coder reliability (confirmability) and thick descriptions of the setting, protocols and transactions were made (transferability). The tool was checked for reliability by perfecting it with the conduct of a pilot at a hospital outside the metropolis. This was to check the; wording, comprehension, reliability and accuracy of the interview guide to elicit the right responses for the phenomena.

## Ethical consideration

We obtained institutional permission from the Tamale Teaching Hospital- Institutional Review Board and the Northern Regional Health Directorate-Tamale first. Ethical clearance was obtained from the Committee on Human Rights, Publication and Ethics (CHRPE) of Kwame Nkrumah University of Science and Technology and Research Development Division of Ghana Health Service with approval numbers (CHRPE/AP/407/19) and (GHS-ERC 051/05/19) respectively in May 2019. This current study is part of a larger project titled "Experiences of Caregivers of Children Living with HIV/AIDS in Tamale Metropolis-Ghana".

After obtaining ethical clearance, the hospital's management and ART clinic in charge were notified before data collection began. During data collection, all participants were duly taken through the full disclosure process before consent by thumb-printing or signing before a witness.

## Results

### Participant information

A total of 9 caregivers were involved in this study, comprising seven (7) females PLWHA and (2) HIV- males. Table 1 below, shows all the seven females PLWHA were caring for their biological child while the males were caring for family members other than biological. The ages of the caregivers range from 24–48 years, with an average of 38. Four (4) of them were not engaged in any economic activity, three (3) were involved in self-employment (laundry, seamstress and petty trading) which did not provide regular income while just two (2) were gainfully employed.

Table 2 below shows six of the children were females and three males. Their average age was 9 years (range from 5–12 years) and had been on the program averagely for 4 years. All these children were attending school; nursery (2), primary (5) and Junior High School (2).

### Core themes

Six themes were realised from the analysis of transcribes as shown in Table 3 below. These were; denial of HIV/AID diagnosis, stock-outs and privacy at the clinic, food insecurity and

**Table 1. Participant information.**

| Caregiver | Sex | Age | HIV-status | Occupation | Relationship with child |
|---|---|---|---|---|---|
| Caregiver -1 | Female | 24 | + | Self-employed | Parent |
| Caregiver -2 | Male | 41 | - | Private employment | Uncle |
| Caregiver -3 | Female | 40 | + | Unemployed | Parent |
| Caregiver -4 | Female | 36 | + | Unemployed | Parent |
| Caregiver -5 | Male | 26 | - | Unemployed | Sister |
| Caregiver -6 | Female | 44 | + | Government employee | Parent |
| Caregiver -7 | Female | 48 | + | Unemployed | Parent |
| Caregiver -8 | Female | 42 | + | Self-employed | Parent |
| Caregiver -9 | Female | 37 | + | Self-employed | Parent |

**Table 2. Biographic data of children living with HIV/AIDS.**

| Pseudo name | Age (years) | Sex | Level of education | How long been on ART (years) |
|---|---|---|---|---|
| CLWHA-1 | 5 | Female | Nursery 2 | 1 |
| CLWHA-2 | 9 | Male | Primary 3 | 3 |
| CLWHA-3 | 12 | Female | Junior High School 2 | 6 |
| CLWHA-4 | 12 | Male | Primary 3 | 7 |
| CLWHA-5 | 11 | Female | Primary 6 | 1 |
| CLWHA-6 | 10 | Male | Junior High School 1 | 8 |
| CLWHA-7 | 11 | Female | Primary 5 | 3 |
| CLWHA-8 | 7 | Female | Primary 2 | 4 |
| CLWHA-9 | 5 | Female | Nursery 2 | 3 |

poor support, ignorance and alternative herbal cure, stigma and discrimination, transportation and distance. These perceived barriers caregivers maintained were faced by themselves as well continually but for their determination to ensure their children have good health. These have been expressed in the narrations of the caregivers.

**Theme one: Denial of HIV/AID diagnosis.** Caregivers express the fear surrounding the disease made it difficult for others to accept the diagnosis. Hence, their inability to come to terms with going to the ART clinic for medications. The following are narrations of caregivers;

*"they easily get to a time they think the disease, as to how me in this state so like my late brother's wife like this she just gave up completely"* (Caregiver -2, M, 41 years, HIV-, other)

*"Some don't believe the disease is true and refuse to accept the diagnosis"* (Caregiver -8, F, 42 years, PLWHA, Parent).

**Theme two: Stock-outs and privacy at the clinic.** ART stock-outs were mentioned as a problem with ARVs (especially the pediatric doses) which made the caregivers not adhere to appointments. Aside from one of the sites which had a separate ART clinic away from the facility's main structures, the other sites' ART clinics had privacy issues. In one it was located at the main Out-Patient Department (OPD) and the other at Ante-Natal Clinic (ANC) closer to the hospital administration. These are expressed as the following;

Caregiver -2, (M, 41 years, HIV-, other) lamented,

*"So sometimes they may run-short of the quantity they give them for a duration at the counselling centre, they come home and hang"*

And with privacy concerning the child's late mother, Caregiver-2 explained,

**Table 3. Summary of themes.**

| No. | Theme |
|---|---|
| 1 | Denial of HIV/AID diagnosis |
| 2 | Stock-outs and privacy at the clinic |
| 3 | Busy schedules and Poor Support |
| 4 | Ignorance and Alternative Herbal Cure |
| 5 | Stigma and Discrimination |
| 6 | Transportation and Distance |

**"**As the mother started with (hospital B), before we realized she moved to (Town Z-Hospital) to take, is like (Town Z-Hospital) was more accessible and more private"

Then with his current difficulty that he felt may deter other caregivers he has this to say,

".. *sometimes when I go to the counselling unit to take the drugs if not because am determined and you sit at the waiting side some of them, they don't feel comfortable looking you, see others turning*"

Others recanted also as below for privacy and medication supply problems

"*Also, disrupted medicine supply makes them go and not come back again, due to low stocks so, they should help for the children medicine to always be there enough*" (Caregiver -4, F, 36 years, PLWHA, Parent)

"*Last time I came and they said there is no medicine, and mostly the medicine will be expired and we have to take the expired ones. I have to take the expired medicines since there are no new ones. . . .Others too when they are coming, they think they know some people who may see them*" (Caregiver -9, F, 37 years, PLWHA, Parent)

**Theme three: Busy schedules and poor support.**   Again, this theme borders on the inability of caregivers to make time to come for ART appointments due to busy schedules. It is further compounded by poor support to help take these CLWHA for ART service when the primary caregivers are indisposed, sick or unable to. The following confirms their perceived barriers with busy schedules and poor support.

"*Sometimes they don't have time or for the children, they may not have someone to come for the medication (thus) they may be busy and may not get time to come for the medication. And the child like this one can't come by herself for the medication if not someone who brings her*" (Caregiver -1, F, 24 years, PLWHA, Parent).

"*They look at the time they will go and waste there and the people will speak to them, they don't like, l don't have time*" (Caregiver -2, M, 41 years, HIV-, other)

"*Others are weak and can't walk here or even move around and don't have support at home*" (Caregiver -6, F, 44 years, PLWHA, Parent).

On support to help take CLWHA for appointment a caregiver shared,

"*Some has not informed their family for them to support and maybe coming to the hospital every month, others will ask why they are coming like that and because they don't want to disclose, they will soon stop coming*" (Caregiver -8, F, 42 years, PLWHA, Parent).

**Theme four: Ignorance and alternative herbal cure.**   This theme shows some caregivers have misconceptions and ignorance about the treatment and management modalities of HIV. It then bars them from cultivating healthy health-seeking behaviours but relying on faked alternative herbal care. This becomes a challenge and served as a barrier for CLWHA gaining the benefits of ARVs. They reported;

*"I will say is lack of education, most of them are not educated and they don't know the results of how it will end if they're not taking the medicines"* (Caregiver -3, F, 40 years, PLWHA, Parent).

*"Some caregivers believe those who say they have the cure to the disease so they go there and don't come here that may be what is preventing most of the people. . . . I don't know how someone can convince me to take such herbal medicine. Herbal medicine can't cure this disease"* (Caregiver -9, F, 37 years, PLWHA, Parent).

*"others too, do not know the importance of the medicine"* (Caregiver -4, F, 36 years, PLWHA, Parent).

*"Some I don't know but they feel like is waste of time. Like they don't have time to come for the medications"* (Caregiver -5, M, 41 years, HIV-, other).

**Theme five: Stigma and discrimination.** Many caregivers expressed the concerns of stigma and discrimination as a barrier to some not attending the ART clinics for medication. As such the fear of seeing known faces, being asked why the frequent hospital visits and being labelled were a barrier. The under-listed quotations provide emphasis on this theme.

*"Because they don't want people to know, because of the stigma they don't want to come and meet known faces and then after that they will be treated like an outcast* (Caregiver -6, F, 44 years, PLWHA, Parent).

*"Some of them feel shy to bring the children because they don't want people to know they're sick of this condition for them to make fun of them. Because people will just spread it"* (Caregiver -7, F, 48 years, PLWHA, Parent).

*"Some too, are afraid to come and see others here"* (Caregiver -4, F, 36 years, PLWHA, Parent).

**Theme six: Transportation and distance.** Cost of transport and distance caregivers bemoaned can become a barrier to many caregivers who want to come for the medication for their CLWHA, even if they were willing to adhere. They expressed lamentations about having to walk a long distance to the clinic. Caregivers narrated the following;

*"The problem is money to come here is a big challenge because for me any little that I get I make sure that the month they have given us if even it is coins, I will save it to come" . . .. Also, the distance is sometimes a problem* (Caregiver -3, F, 40 years, PLWHA, Parent).

*"Others too don't have money to take a taxi for the fare. I use to meet people here who complain that they walked here and they don't have the support to come here"* (Caregiver -6, F, 44 years, PLWHA, Parent)

*"Sometimes the money for transport is a problem sometimes. For me sometimes I hold my bag and walk here with her"* (Caregiver -9, F, 37 years, PLWHA, Parent)

## Discussion

Our study explored perceived barriers to ART enrollment and adherence for caregivers of children living with HIV/AIDS. The study enquired from caregivers who were visiting the clinics with their children for treatment why others are unable to honour their appointments and also

ensure their children adhere to the regimen. The adherence of CLWHA to ART appointments ensures continual monitoring, decreases viral resistance and viral load, and positively impacts the quality of life. Caregivers in this study underscored the importance of adhering to ARVs however, expressed barriers deter others from coming for treatment. They further explained they also do go through some of these problems in a bid to ensure their children get their medications.

In this study caregivers identified the following as barriers to CLWHA ART enrollment and adherence; denial of HIV/AID diagnosis, stock-outs and privacy at the clinic, busy schedule and poor support, ignorance and alternative herbal cure, stigma and discrimination, cost of transportation long-distance dance. The current study is largely consistent with barriers identified by PLWHA ART initiation in Kenya [29] on stigma, denial of diagnosis and difficulty in obtaining refills of ARVs(stock out) but not side effects of the medications. The similarity is because of the inadequate supply of ARVs as well as fear and misconceptions about HIV/AIDS in Africa. However, because CLWHA is unable to express themselves well on the side effects as compared to adults' caregivers did not mention side effects as a barrier to enrollment and adherence in this study. It is therefore important for the government to ensure an adequate supply of ART stocks for PLWA, especially pediatric doses. The use of expired ARVs in this study shows the precarious nature of stock-outs in many African settings and calls for conscious efforts to desist from the act.

Caregivers' expression of denial or non-acceptance of HIV status and stigma in this study also concord with a significant barrier among non-adherent adolescents living with HIV in Botswana [18]. However, in contrast to this current study food insecurity and side effects of the medications were also reported as barriers in Kenya. The denial of diagnosis shows a grave concern for HIV care as adherence to interventions becomes a serious threat. This undermines the health-seeking behaviour of caregivers for children. Re-enforcement of adherence counselling is pivotal to curb this worrying scenario.

Findings also reflect PLWH in rural Zambia [30] and Gaza [31] where caregivers experienced barriers to clinic attendance relating to time and distance to the ART clinic and the cost of transportation. This similarity explains the less accessibility of PLWH to ART centres. Even though the treatment is free in Ghana, PLWH has to move long distances to access the treatment. Especially in this metropolis where only three facilities provide ART Services. As other facilities do the counselling and testing and refer to these three facilities making PLWH move long distances to access ARVs. As well due to work, sickness and other roles caregivers engages in its difficult to get time to attend ART appointment. However, this study is counter-intuitive to a study in rural Gaza [31] where lack of confidence in the national health system was mentioned as a barrier. The difference, however, could be because this current study was conducted in an urban area.

Maccarthy et al. identified the inability to buy food, the burden of taking multiple medications, school attendance, and limiting privacy as barriers in Uganda among adolescents living with HIV/AIDS which contrast this current study. Caregivers in this study did not see pill burden and school attendance as barriers and provide great relief as many PLWHA are not citing this as a barrier. But agrees with poor family support due to unreliable constant change in guardianship because they had lost their biological parents to HIV. With many caregivers keeping CLWHA status a secret, it is difficult for other people to help with clinic appointments and ARV administration in their absence. This calls for collective deliberations and support for home care that involves significant others of CHLWA.

The current study also contrasts a study conducted in Vietnam [32] where there exist poor linkage between HIV voluntary testing and care and treatment services, poor confidentiality and inadequate HIV/AIDS specialist were identified as structural barriers to ART initiation.

The current study does not have confidentiality and poor coordination of testing and treatment as a concern. Health care providers need to maintain their positive conduct to PLWHA, to ensure PLWHA feel positive about coming to the clinics for medication, adherence counselling and health education.

Familiarity is found in this study to believe in alternative herbal cures but not fear of side effects of treatment and substance misuse identified in rural KwaZulu-Natal, South Africa [17]. Where it was identified, many were testing knowing their status but not enrolling on the ART program to treat. Belief in alternative herbal and faith practitioners is common in African settings and possess a serious threat to HIV interventions. The need to extend education to these religious leaders, priests, herbal and alternative herbal practitioners to demystify HIV cure is pivotal. Substance abuse was not mentioned in this study as a barrier probably because a majority of the caregivers were females. As has been reported in a previous survey conducted in Low and middle-income countries where alcoholism was strongly associated with the male gender [33]. Also, the setting of this current study is a predominantly Muslim society where the abuse of substances such as alcohol and illicit drugs is frowned upon.

Our current study is familiar with [23] in Southern Ghana where barriers to ART centred on the high financial burden associated with accessing and receiving ART, shortage of ARVs and treatment of opportunistic infections, stigma, and long-distance to treatment centres. However, inconsistent with barriers such as delays associated with receiving care from the treatment centres, fear of side effects of ARVs, and job insecurity arising from a regular leave of absence to receive ART. This similarity shows an in-country indirect cost for ART appointments, stigma and frequent shortage of ARVs. It creates a difficult situation for proper adherence for PLWH in Ghana, especially for dependent children. Differences in job security come as a result of many of the caregivers being unemployed or undertaking self-jobs. There were no complaints of long queues and stay at the hospital due to the low prevalence of HIV/AIDS in Northern Ghana compare to the south.

Ankomah et al. [34] corroborate the position of this current study where the mere presence of a person at the HIV counselling clinic is enough for the person to be labelled as or suspected to be an HIV patient. Indicating a high perception of stigmatization which serve as a barrier to ART. The study further contrasts their finding of PLWH citing the quality of care given by some health workers and conducts that breached confidentiality about their clients' health status as barriers in this metropolis. This may compel many patients and potential users of ART clinic services not to patronize services.

We recommend that future studies explore barriers from non-adherent CLWHA caregivers. Active involvement of significant others in-clinic appointments by care providers is highly recommended in this study. Health care facilities together with the ministry should ensure adequate stocks of ARVs. Faith and alternative herbal practitioners should be educated to demystify notions of curing HIV. Incentives for adherent mothers as a way of motivation for other caregivers will be a significant factor to reduce the attrition of CLWHA on ART. We also recommend home visits as crucial to medication adherence. During these visits the stocks of medications could be replenished, health education given and sound interventions made based on an assessment of the home environment. Likewise, telehealth interventions such as sending mobile messages as reminders for appointments and information on ARVs are highly suggested.

## Strength and limitations

Our study, being the first to explore such an important aspect of CLWHA intervention allowed for starting of a discourse that will improve their health outcomes. The approach of the study

allowed for deep-seated narrations of the perceived barriers caregivers goes through to adhere to ART appointment.

Our study is limited by language (English and Daghani), exclusion of caregivers of children who were not on ART and the inability to collect views of the care providers about the possible barriers as well. Despite these limitations, the findings of this study are valid for the ongoing discussion on the barriers to ART.

## Conclusion

Ghana among its sister countries in Africa has strived to improve health coverage with the implementation of the CHPS program. Affordability of health care to vulnerable groups such as children, women and the elderly has been at the centre of health policy. Introduction of free maternal health care, National Health Insurance Scheme and among provident interventions in making health affordable to citizens. Since the first case of HIV was reported in this republic, the government has ensured free voluntary screening and testing of HIV/AIDS, enrollment in ART, and prevention campaigns among others. Yet there exist multi-dimensional barriers to ART in Ghana, especially among hard-to-reach populations like children.

Perceived barriers to ART and adherence are multifaceted, encompassing caregiver, child, healthcare provider and facilities factors. Caregivers recognize the ultimate importance of honouring clinic appointments and the improvement in CLHA health with ARVs. They profess to encounter perceived barriers others face. These barriers border on; denial of diagnosis, stock out, poor privacy, cost of transportation, long distances to the clinics, stigma, busy schedules, poor support, ignorance of the condition and belief in alternative herbal medicines. They can navigate these challenges due to resilience and need to ensure their CLWHA are healthy.

## Supporting information

**S1 File. Interview guide.**
(DOCX)

**S2 File. Transcript.**
(DOCX)

## Acknowledgments

We are highly indebted to the participants of this study for their patience to provide worthwhile information for the study. We extend our appreciation to the staff and management of the hospitals and ART/STI clinic management. As well our heartfelt thanks to the language experts for their immense contribution to the success of this study.

## Author Contributions

**Conceptualization:** Gideon Awenabisa Atanuriba.

**Data curation:** Gideon Awenabisa Atanuriba, Timothy Gazari, Timothy Tienbia Laari.

**Formal analysis:** Gideon Awenabisa Atanuriba, Timothy Gazari, Timothy Tienbia Laari.

**Investigation:** Gideon Awenabisa Atanuriba, Timothy Tienbia Laari.

**Methodology:** Gideon Awenabisa Atanuriba, Felix Apiribu, Philemon Adoliwine Amooba, Adwoa Bemah Boamah Mensah, Richard Adongo Afaya.

**Project administration:** Felix Apiribu, Veronica Millicent Dzomeku.

**Resources:** Gideon Awenabisa Atanuriba, Felix Apiribu, Richard Adongo Afaya, Timothy Tienbia Laari, Moses Haruna Akor.

**Supervision:** Felix Apiribu, Veronica Millicent Dzomeku, Philemon Adoliwine Amooba, Adwoa Bemah Boamah Mensah.

**Validation:** Felix Apiribu, Richard Adongo Afaya, Timothy Gazari, Timothy Tienbia Laari.

**Visualization:** Timothy Gazari.

**Writing – original draft:** Gideon Awenabisa Atanuriba, Timothy Tienbia Laari, Moses Haruna Akor.

**Writing – review & editing:** Felix Apiribu, Philemon Adoliwine Amooba, Richard Adongo Afaya, Timothy Gazari, Linda Abnory.

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
