## [Decision Letter · Decision Letter 0]

3 Dec 2021

PONE-D-21-21063Some Believe those who say they can Cure it” Perceived Barriers to Antiretroviral Therapy for Children Living with HIV/AIDS: Qualitative Exploration of Caregivers Experiences in Tamale MetropolisPLOS ONE

Dear Dr. Atanuriba,

Thank you for submitting your manuscript to PLOS ONE. After careful consideration, we feel that it has merit but does not fully meet PLOS ONE’s publication criteria as it currently stands. Therefore, we invite you to submit a revised version of the manuscript that addresses the points raised during the review process.

Firstly we would like to apologies for the delays faced on your submission and  thank you for your patience.

The manuscript has been evaluated by two reviewers, and their comments are available below. The reviewers have requested some additional  points for further clarification. In particular they requested information on the rational behind the approach used for qualitative interviewing, as well as details on how data saturation was determined. 

Finally, the reviewers feel that copy editing can further improve the manuscript. In particular please amend your manuscript to adhere to our submission guidelines with respect to language describing demographic groups. Outmoded terms and potentially stigmatizing labels should be changed to more current, acceptable terminology. Specifically, we recommend that “HIV positive” should be changed to more appropriate term(s). Please note that PLOS ONE cannot provide copyediting for manuscripts.

Could you please carefully revise the manuscript to address all comments raised?

We look forward to receiving your revised manuscript.

Kind regards,

Lucinda Shen, MSc

Staff Editor

PLOS ONE

Journal Requirements:

2. Please include a copy of the interview guide used in the study, in both the original language and English, as Supporting Information, or include a citation if it has been published previously.

Reviewers' comments:

Reviewer's Responses to Questions

**Comments to the Author**

1. Is the manuscript technically sound, and do the data support the conclusions?

Reviewer #1: Partly

Reviewer #2: Yes

2. Has the statistical analysis been performed appropriately and rigorously? 

Reviewer #1: Yes

Reviewer #2: N/A

3. Have the authors made all data underlying the findings in their manuscript fully available?

Reviewer #1: Yes

Reviewer #2: Yes

4. Is the manuscript presented in an intelligible fashion and written in standard English?

Reviewer #1: No

Reviewer #2: Yes

5. Review Comments to the Author

Reviewer #1: The authors use a qualitative exploration to understand the barriers to ART enrolment and adherence among children living with HIV/AIDS in the Tamale Metropolis in the era of the ‘treat all’ policy, from the perspectives of the caregivers. Denial of HIV/AIDS diagnosis, stock-outs and privacy at the clinic, busy schedule and poor support, ignorance and alternative herbal cure, stigma and discrimination, and transportation and distance emerged as barriers to ART enrolment and adherence.

Findings from the study provide useful insight into the barriers to ART adherence among children that could be addressed, especially, at the health facility level to improve adherent to therapy among children living with HIV.

However, I have some major concerns about the research methodology, results, discussion and conclusions that need to be addressed before the manuscript can be published:

Methods

1. On page 9, the authors state, ‘Owing to the varied biographic background (age, sex, and religion) of the caregivers which did not favour focused group discussion’, as a reason for opting for in-depth interviews. However, focus group discussions (FGDs) can also be categorised according to the different participants’ characteristics to achieve a homogeneous group for a balanced discussion. By itself, this may not be a justifiable reason for choosing in-depth interviews over FGDs, and the authors should clarify if there were any other reasons besides that.

2. Could the authors give more details about how data saturation was determined. Was iteration considered in the data collection and analysis to guide this process?

Results

3. The term ‘HIV positive’ is generally considered stigmatizing and ‘people living with HIV’ is preferred. The authors should edit this throughout the manuscript.

4. From the illustrative quotes presented, stock-outs and privacy at the clinic relate to two different barriers. Why did the authors choose to organise these under the same theme?

5. On page 15, under theme two, the authors report,

‘Aside one of the sites which has a separate ART clinic away from the facility main structures, the other sites ART clinic had privacy issues. In one it was located at the main Out-Patient Department (OPD) and the other at Ante-Natal Clinic (ANC) closer to the hospital administration.’

Was this reported by the caregivers or was it an assessment made by the authors through observation? Data collector observation may not necessarily reflect the perceptions of the caregivers and is not covered under the stated data collection methods.

Discussion

6. On page 21, Paragraph 1, the authors make the conclusion that because confidentiality and poor coordination of testing and treatment were not identified as barriers in this study, unlike findings in previous studies, ‘there exist properly trained counsellors and HIV professional providing services’ and that ‘This contrast underpins better professional ethics and conduct of care providers in this metropolis.’ However, this assumption may not hold and it is not supported by findings of the study.

7. In paragraph 2, the authors also write that, ‘Substance abuse was not mentioned in this study as a barrier because a majority of the caregivers were females.’ It may be useful to explain how gender is related to alcohol use and present supporting references.

Strengths and limitations

8. The authors pose sample size as a limitation of the study. However, if sampling was done until data saturation was reached, as reported, this ought not to be a concern. How do the authors think having a larger sample size would have improved the study?

9. Wouldn’t it have been useful to also include caregivers of children who were not on ART to explore barriers to ART enrolment? The authors could consider including this as a limitation of the study.

Conclusions

10. In the last statement, the authors conclude that, the caregivers ‘are able to navigate these challenges due to resilience and need to ensure their CLWHA are healthy’. This however does not seem derived from the study findings.

There were also a few minor issues:

Background

1. Some references for statistics cited are not provided for example in the last paragraph on Page 4 and first paragraph on page 5.

Methods

2. Were the caregivers required to speak both English and Daghani or either of the two. Please clarify on Page 8.

3. On page 9, was the presence of a witness during the consenting process a requirement for all participants, including those who could read and write?

Results

4. For Tables 1 and 2:

a) It may be better to write in full female and male, instead of F and M, or give a key at the bottom of the table

b) Unemployed may be a better term to use than ‘Nill’ under occupation.

c) It would be good to specify what the ‘other’ under ‘relationship to the child’ is for.

5. The authors in the first paragraph on Page 12 write that, ‘three (3) [of the caregivers] involved in self-employment (laundry, seamstress and petty trading) which does not provide regular income’. Was the conclusion that these occupations do not provide regular income made by the authors based on the nature of the work or was this self-reported by the caregivers?

Discussion

6. Some of the referencing needs formatting, for example, in the first sentence in paragraph 1 on page 22.

7. Some statements are incomplete, for example, the second last paragraph on page 22, ‘Active involvement of significant others in clinic appointment by care providers.’

Generally, language editing is recommended for the entire manuscript to improve readability.

Reviewer #2: Review

Overall

Great study, very interesting findings. I think the part about herbal / traditional is very important in the Ghanian context, so I found that particularly interesting.

Some of the sentences could be restructured to be more concise or clear. For example, the minutes of interviews were 47, this would be better structured as “The interviews lasted an average of X minutes.” which is more active and clearer. I would recommend doing a read outloud to find small structure issues like this.

I would like to understand better what is unique about this study compared to other similar studies. I would like a better understanding of the context and population and how this study makes a unique contribution. This is not quite clear to me now, but there is much potential here. I think expanding on Tamale and the population would help.

Did religion come up at all? I know Northern Ghana is largely muslim so I wondered if any sort of religious factors arose?

Abstract

This is minor but there are some typos in the abstract which take away from the quality of the work. For example, “there exist” should be “there exists”.

“HIV and AIDS” is later referred to as “HIV/AID”. I would make this consistent throughout. The abstract is a bit long/unbalanced-- the background is very long whereas the results are short.

Ethics Statement

Do not need to say the second “respectively”.

Title

I would remove the capitalizations from Cure and Believe in the title, as it is a quote. It would potentially be better to be consistent about capitalization in the whole title.

Introduction

Pick HIV and AIDS rather than saying it that way sometimes, and other times using HIV/AIDS. I would recommend the latter.

Some citations are missing in the first few sentences of the introduction. For example, for “as many more infected are having their quality and expectancy of life improving.” “diseases of history have exerted systematic and structural effects”.

My same comment about sentence structure is relevant in the introduction. For example, the sentence “Evidence from Ghana shows CLWHA (0–14 years) are about 25,955(8%) with an estimated 2,972(15%) new infections and 2,441 (18%) deaths.” should be something more like, “In Ghana, there are 25,955 0-14 year olds (8%)...” to make it more active and clear.

Methods

“Fathered by Husserl the meaning and essence people have about a phenomenon is described while making sure presuppositions are bracketed(25)” I don’t understand this sentence and generally the first paragraph. Can it be explained in a less jargony fashion?

First sentence about Tamale is missing a citation

Which hospitals were selected? Name them

The sentence “The hospitals which serve as training centers for students, provides in-patient and out-patient care services” should be re-phrased. “The hospital has a training center for student, inpatient care services, and outpatient care services.”

“Their exposition of the phenomena perceived barriers because they were asked, what are some of the reasons that prevent other caregivers from bringing their CLWHA to the ART clinics for treatment?” Is this the only question that was asked? Could you include the questionnaire with the paper? This sentence seems out of place given the Instrument Section exists.

The steps of Colaizi could be better presented in a figure or a chart of some kind rather than in a dense paragraph.

Add citations for Colaizi and Lincoln and Guba

Don't understand this sentence please rephrase “Haven obtained ethical clearance the hospital management and ART clinic in charges were notified before data collection began”

Results

Good summary of participants!

I like the presentation of themes

Discussion

These sentences are missing something in the beginning, I guess an author name: “(15) identified inability to buy food, the burden of taking multiple medications and school attendance limiting privacy as barriers in Uganda among adolescence living with HIV/AIDS which contrast this current study” “33) corroborates the position of this current study when it was noted that the mere presence of a person at the HIV counselling clinic is enough for the person to be labelled as or suspected to be an HIV patient. Indicating a high perception of stigmatization which serve as a barrier to ART.”

Some of the studies it is compared to could be summarized, rather than a separate sentence about all of them

The last paragraph before strengths and limitations could be expanded, especially if the other literature is summarized a bit more concisely. I would expand on these recommendations and consider grounding them in other studies which have shown these suggestions have been successful elsewhere.

6. PLOS authors have the option to publish the peer review history of their article (what does this mean?). If published, this will include your full peer review and any attached files.

Reviewer #1: No

Reviewer #2: **Yes: **Aldina Mesic

---

## [Author Response · Author response to Decision Letter 0]

31 Mar 2022

“SOME BELIEVE THOSE WHO SAY THEY CAN CURE IT” PERCEIVED BARRIERS TO ANTIRETROVIRAL THERAPY FOR CHILDREN LIVING WITH HIV/AIDS: QUALITATIVE EXPLORATION OF CAREGIVERS EXPERIENCES IN TAMALE METROPOLIS

Following reception of reviewers comment of the above, the authors provides the understated response

Reviewer 1

Comment: n page 9, the authors state, ‘Owing to the varied biographic background (age, sex, and religion) of the caregivers which did not favour focused group discussion’, as a reason for opting for in-depth interviews. However, focus group discussions (FGDs) can also be categorised according to the different participants’ characteristics to achieve a homogeneous group for a balanced discussion. By itself, this may not be a justifiable reason for choosing in-depth interviews over FGDs, and the authors should clarify if there were any other reasons besides that

Response: Comment acknowledged and further explanations provided 

comment: Could the authors give more details about how data saturation was determined. Was iteration considered in the data collection and analysis to guide this process?

response: Yes, iteration was considered and we have provided further information on saturation

comment: The term ‘HIV positive’ is generally considered stigmatizing and ‘people living with HIV’ is preferred. The authors should edit this throughout the manuscript

response: Comment acknowledged and corrections made 

comment: From the illustrative quotes presented, stock-outs and privacy at the clinic relate to two different barriers. Why did the authors choose to organise these under the same theme?

response: We recognized both barriers as related to facility/hospital level barriers. 

As such put them together

comment: On page 15, under theme two, the authors report,

‘Aside one of the sites which has a separate ART clinic away from the facility main structures, the other sites ART clinic had privacy issues. In one it was located at the main Out-Patient Department (OPD) and the other at Ante-Natal Clinic (ANC) closer to the hospital administration.’

Was this reported by the caregivers or was it an assessment made by the authors through observation? Data collector observation may not necessarily reflect the perceptions of the caregivers and is not covered under the stated data collection methods.

response: Comment acknowledged however, caregivers narrated how the proximity of the clinics to OPD and administration made it easy for others to identify them as PLWHA during our final step of data validation (member checks) 

The introduction of adding such information is to make it much clearer for readers. 

comment: On page 21, Paragraph 1, the authors make the conclusion that because confidentiality and poor coordination of testing and treatment were not identified as barriers in this study, unlike findings in previous studies, ‘there exist properly trained counsellors and HIV professional providing services’ and that ‘This contrast underpins better professional ethics and conduct of care providers in this metropolis.’ However, this assumption may not hold and it is not supported by findings of the study.

response: Upon a careful consideration, this portion has been deleted from the manuscript.

comment: In paragraph 2, the authors also write that, ‘Substance abuse was not mentioned in this study as a barrier because a majority of the caregivers were females.’ It may be useful to explain how gender is related to alcohol use and present supporting references.

response: The relationship between gender and alcohol abuse has been explained and referenced

comment: The authors pose sample size as a limitation of the study. However, if sampling was done until data saturation was reached, as reported, this ought not to be a concern. How do the authors think having a larger sample size would have improved the study?

response: We accept the comment and do, delete same, Thank you 

comment: Wouldn’t it have been useful to also include caregivers of children who were not on ART to explore barriers to ART enrolment? The authors could consider including this as a limitation of the study.

response: Suggestion accepted and added to the limitations 

comment: In the last statement, the authors conclude that, the caregivers ‘are able to navigate these challenges due to resilience and need to ensure their CLWHA are healthy’. This however does not seem derived from the study findings

response: The nature of these perceived barriers shows these caregivers equally faces same. Its therefore their resolve to adhere to medications that makes them navigate these barriers 

comment: Some references for statistics cited are not provided for example in the last paragraph on Page 4 and first paragraph on page 5.

response: The said references have been provided. 

comment: Were the caregivers required to speak both English and Daghani or either of the two. Please clarify on Page 8.

response: They were required to speak either of the two. Amends made.

comment: On page 9, was the presence of a witness during the consenting process a requirement for all participants, including those who could read and write?

response: Yes, please it was required for all 

comment: It may be better to write in full female and male, instead of F and M, or give a key at the bottom of the table. Unemployed may be a better term to use than ‘Nill’ under occupation, It would be good to specify what the ‘other’ under ‘relationship to the child’ is for

response: Comment accepted and corrections made 

comment: The authors in the first paragraph on Page 12 write that, ‘three (3) [of the caregivers] involved in self-employment (laundry, seamstress and petty trading) which does not provide regular income’. Was the conclusion that these occupations do not provide regular income made by the authors based on the nature of the work or was this self-reported by the caregivers?

response: Please it was self-reported 

comment: Some of the referencing needs formatting, for example, in the first sentence in paragraph 1 on page 22

response: Reference formatted as suggested 

comment: Some statements are incomplete, for example, the second last paragraph on page 22, ‘Active involvement of significant others in clinic appointment by care providers.’

Generally, language editing is recommended for the entire manuscript to improve readability.

response: Comment acknowledged 

Sentence completed 

 reviewer 2 

comment: Some of the sentences could be restructured to be more concise or clear. For example, the minutes of interviews were 47, this would be better structured as “The interviews lasted an average of X minutes.” which is more active and clearer. I would recommend doing a read outloud to find small structure issues like this.

response: Language and other editing has been made throughout the entire script

comment: I would like to understand better what is unique about this study compared to other similar studies.

response: Our study is unique compared to others, because it explored barriers from caregivers who had enrolled their children on ART and were adhering to appointments. Thus enquiring from them why other caregivers were not enrolling and/or adhering to appointment. 

This is taken from the context that in ghana ART coverage is 46.55% among adults (15+ years), 53.76% women (15+ years) and just 25.55% children (0-14years). 

For children living with HIV/AIDS their caregivers are the most critical people in their lives since they are dependent. 

Our study also contrast other studies that explores barriers to ART among adults 

 comment: I would like a better understanding of the context and population and how this study makes a unique contribution

response: Context is taken from the premise that in Ghana ART coverage is 46.55% among adults (15+ years), 53.76% women (15+ years) and just 25.55% children (0-14years). 

As such nearly a seventh of children living with HIV/AIDS are not enrolled unto ART compared to a much better statistic for adults and women. 

Yet, the power of making this very important decision lies in the hands of adults. 

To this end since ability to recruit caregivers of children living with HIV/AIDS not enrolled and /or adherent was difficult, we approached caregivers of adherent children. This was to explore their “perceived” reasons why other caregivers refuse to enroll their children and /or adhere to appointments. 

The findings provide unique knowledge about barriers to ART which are also confronted by adherent caregivers. Thereby providing provident solutions that are useful for both categories of caregivers in the metropolis. 

comment: This is not quite clear to me now, but there is much potential here. I think expanding on Tamale and the population would help.

response: Indeed, we recognize so 

But in this study we purposively selected facilities with ART clinics for this study. 

comment: Did religion come up at all? I know Northern Ghana is largely muslim so I wondered if any sort of religious factors arose?

response: Yes, it did come up 

But was not strongly attached to any of the faiths as reviewer sought to know 

comment: This is minor but there are some typos in the abstract which take away from the quality of the work. For example, “there exist” should be “there exists”.

“HIV and AIDS” is later referred to as “HIV/AID”. I would make this consistent throughout. The abstract is a bit long/unbalanced-- the background is very long whereas the results are short.

response: Thank you for your comment 

Amends has been made 

comment: Do not need to say the second “respectively”. I would remove the capitalizations from Cure and Believe in the title, as it is a quote. It would potentially be better to be consistent about capitalization in the whole title. Pick HIV and AIDS rather than saying it that way sometimes, and other times using HIV/AIDS. I would recommend the latter.

response: Comment acknowledged and corrections made

comment: Some citations are missing in the first few sentences of the introduction. For example, for “as many more infected are having their quality and expectancy of life improving.” “diseases of history have exerted systematic and structural effects”.

response: References has been provided 

comment: My same comment about sentence structure is relevant in the introduction. For example, the sentence “Evidence from Ghana shows CLWHA (0–14 years) are about 25,955(8%) with an estimated 2,972(15%) new infections and 2,441 (18%) deaths.” should be something more like, “In Ghana, there are 25,955 0-14 year olds (8%)...” to make it more active and clear. 

Fathered by Husserl the meaning and essence people have about a phenomenon is described while making sure presuppositions are bracketed(25)” I don’t understand this sentence and generally the first paragraph. Can it be explained in a less jargony fashion?

response: Comment acknowledged and amends made

comment: First sentence about Tamale is missing a citation

response: Reference provided 

comment: Which hospitals were selected? Name them

response: Amends made and hospitals named 

comment: The sentence “The hospitals which serve as training centers for students, provides in-patient and out-patient care services” should be re-phrased. “The hospital has a training center for student, inpatient care services, and outpatient care services.”

response: Comment acknowledged and amends made

comment: Their exposition of the phenomena perceived barriers because they were asked, what are some of the reasons that prevent other caregivers from bringing their CLWHA to the ART clinics for treatment?” Is this the only question that was asked? Could you include the questionnaire with the paper? This sentence seems out of place given the Instrument Section exists.

response: The interview guide has been included 

And some of the probes added to the text 

comment: The steps of Colaizi could be better presented in a figure or a chart of some kind rather than in a dense paragraph.

response: Comment acknowledged and the sentences has been put into a figure (1)

comment: Add citations for Colaizi and Lincoln and Guba

response: Reference provided

comment; Don't understand this sentence please rephrase “Haven obtained ethical clearance the hospital management and ART clinic in charges were notified before data collection began”

These sentences are missing something in the beginning, I guess an author name: “(15) identified inability to buy food, the burden of taking multiple medications and school attendance limiting privacy as barriers in Uganda among adolescence living with HIV/AIDS which contrast this current study” “33) corroborates the position of this current study when it was noted that the mere presence of a person at the HIV counselling clinic is enough for the person to be labelled as or suspected to be an HIV patient. Indicating a high perception of stigmatization which serve as a barrier to ART.”

Some of the studies it is compared to could be summarized, rather than a separate sentence about all of them

The last paragraph before strengths and limitations could be expanded, especially if the other literature is summarized a bit more concisely.

response: Comment acknowledged and amends made 

comment: I would expand on these recommendations and consider grounding them in other studies which have shown these suggestions have been successful elsewhere

response; The section has been reviewed and more recommendations added base on literature. 

Yours sincerely 

Gideon Awenabisa Atanuriba 

atanuriba@gmail.com

+233541186103

---

## [Decision Letter · Decision Letter 1]

27 May 2022

PONE-D-21-21063R1Some Believe those who say they can Cure it” Perceived Barriers to Antiretroviral Therapy for Children Living with HIV/AIDS: Qualitative Exploration of Caregivers Experiences in Tamale MetropolisPLOS ONE

Dear Dr. Atanuriba,

Thank you for submitting your manuscript to PLOS ONE. After careful consideration, we feel that it has merit but does not fully meet PLOS ONE’s publication criteria as it currently stands. Therefore, we invite you to submit a revised version of the manuscript that addresses the points raised during the review process.

The reviewer requests further clarifications in the discussion, and requests that the manuscript is further copy edited to improve the language quality and readability. 

We look forward to receiving your revised manuscript.

Kind regards,

Jamie Royle

Staff Editor

PLOS ONE

Journal Requirements:

Reviewers' comments:

Reviewer's Responses to Questions

**Comments to the Author**

1. If the authors have adequately addressed your comments raised in a previous round of review and you feel that this manuscript is now acceptable for publication, you may indicate that here to bypass the “Comments to the Author” section, enter your conflict of interest statement in the “Confidential to Editor” section, and submit your "Accept" recommendation.

Reviewer #1: (No Response)

2. Is the manuscript technically sound, and do the data support the conclusions?

Reviewer #1: Yes

3. Has the statistical analysis been performed appropriately and rigorously? 

Reviewer #1: Yes

4. Have the authors made all data underlying the findings in their manuscript fully available?

Reviewer #1: Yes

5. Is the manuscript presented in an intelligible fashion and written in standard English?

Reviewer #1: No

6. Review Comments to the Author

Reviewer #1: The authors have provided adequate responses to most of the queries and comments.

I acknowledge the authors' response to the comment in the discussion section on page 23, paragraph 1. I however suggest that the statement ‘Substance abuse was not mentioned in this study as a barrier because a majority of the caregivers were females’ is changed to ‘Substance abuse was not mentioned in this study as a barrier PROBABLY because a majority of the caregivers were females’ since this is not something stemming from the data. It is merely an assumption the authors are making basing on previous studies.

I also recommend that the authors review the manuscript for language, especially in the methods section and discussion, to improve readability.

7. PLOS authors have the option to publish the peer review history of their article (what does this mean?). If published, this will include your full peer review and any attached files.

Reviewer #1: **Yes: **Jacquellyn Nambi Ssanyu

---

## [Author Response · Author response to Decision Letter 1]

30 Jun 2022

Tamale Central Hospital,

Pediatric Department, 

Post Office Box TL2649,

Ghana,

Northern Region,

Tamale.

30 June, 2022

The Editor, 

PLOS ONE Journal,

Dear Sir,

AUTHORS RESPONSE LETTER TO RESEARCH ARTICLE TITLED

 “SOME BELIEVE THOSE WHO SAY THEY CAN CURE IT” PERCEIVED BARRIERS TO ANTIRETROVIRAL THERAPY FOR CHILDREN LIVING WITH HIV/AIDS: QUALITATIVE EXPLORATION OF CAREGIVERS EXPERIENCES IN TAMALE METROPOLIS

We the authors of the above research article under your review, haven received comments for revision do hereby provide the following responses.

Reviewer Comment Page Authors response

Reviewer 1

I acknowledge the authors' response to the comment in the discussion section on page 23, paragraph 1. I however suggest that the statement ‘Substance abuse was not mentioned in this study as a barrier because a majority of the caregivers were females’ is changed to ‘Substance abuse was not mentioned in this study as a barrier PROBABLY because a majority of the caregivers were females’ since this is not something stemming from the data. It is merely an assumption the authors are making basing on previous studies. 23 Comment acknowledged and revision made as suggested. 

I also recommend that the authors review the manuscript for language, especially in the methods section and discussion, to improve readability. Throughout the script Thank you for your comment 

Amends has been made.

As the script has been carefully read for language and editing to ensure readability. 

We also bring to your noticed that none of cited papers have been retracted. 

Given the time this manuscript has been under review; we humbly request for an expedited action for publication. 

 Is our hope this meet your kind consideration and action.

Thank you 

Yours sincerely 

Gideon Awenabisa Atanuriba 

atanuriba@gmail.com

+233541186103

---

## [Decision Letter · Decision Letter 2]

19 Sep 2022

Some Believe those who say they can Cure it” Perceived Barriers to Antiretroviral Therapy for Children Living with HIV/AIDS: Qualitative Exploration of Caregivers Experiences in Tamale Metropolis

PONE-D-21-21063R2

Dear Dr. Atanuriba,

We’re pleased to inform you that your manuscript has been judged scientifically suitable for publication and will be formally accepted for publication once it meets all outstanding technical requirements.

Kind regards,

AbdulAzeez Adeyemi Anjorin, Ph.D.

Academic Editor

PLOS ONE

Additional Editor Comments (optional):

Reviewers' comments:

Reviewer's Responses to Questions

**Comments to the Author**

1. If the authors have adequately addressed your comments raised in a previous round of review and you feel that this manuscript is now acceptable for publication, you may indicate that here to bypass the “Comments to the Author” section, enter your conflict of interest statement in the “Confidential to Editor” section, and submit your "Accept" recommendation.

Reviewer #1: All comments have been addressed

2. Is the manuscript technically sound, and do the data support the conclusions?

Reviewer #1: Yes

3. Has the statistical analysis been performed appropriately and rigorously? 

Reviewer #1: Yes

4. Have the authors made all data underlying the findings in their manuscript fully available?

Reviewer #1: Yes

5. Is the manuscript presented in an intelligible fashion and written in standard English?

Reviewer #1: Yes

6. Review Comments to the Author

Reviewer #1: I thank the authors for the revision and work on the manuscript. I have no further comments for them.

7. PLOS authors have the option to publish the peer review history of their article (what does this mean?). If published, this will include your full peer review and any attached files.

Reviewer #1: **Yes: **Jacquellyn Nambi Ssanyu

---

## [Editor Report · Acceptance letter]

22 Sep 2022

PONE-D-21-21063R2 

  “Some believe those who say they can cure it” Perceived Barriers to Antiretroviral Therapy for Children Living with HIV/AIDS: Qualitative Exploration of Caregivers Experiences in Tamale Metropolis 

Dear Dr. Atanuriba:

I'm pleased to inform you that your manuscript has been deemed suitable for publication in PLOS ONE. Congratulations! Your manuscript is now with our production department. 

Kind regards, 

on behalf of

Dr. AbdulAzeez Adeyemi Anjorin 

Academic Editor

PLOS ONE